

# Novel assessment of numerical forecasting model relative humidity with satellite probabilistic estimates

Chloé Radice[1], Hélène Brogniez[1], Pierre-Emmanuel Kirstetter[2,3], Philippe Chambon[4]

[1] Université Paris-Saclay, UVSQ, CNRS, LATMOS/IPSL, 78280, Guyancourt, France.
[2] University of Oklahoma, Norman, Oklahoma, USA
[3] NOAA/National Severe Storms Laboratory, Norman, Oklahoma
[4] CNRM, Université de Toulouse, Météo France, CNRS, Toulouse, France

*Correspondence to*: Chloé Radice (chloe.radice@latmos.ipsl.fr)

**Abstract.**

A novel method of comparison between an atmospheric model and satellite probabilistic estimates of relative humidity (RH) in the tropical atmosphere is presented. The method is developed to assess the Météo-France numerical weather forecasting model ARPEGE using probability density functions (PDF) of RH estimated from the SAPHIR microwave sounder. The

satellite RH reference is derived by aggregating footprint-scale probabilistic RH to match the spatial and temporal resolution of ARPEGE over the April-May-June 2018 period. The probabilistic comparison is discussed with respect to a classical deterministic comparison confronting each model RH value to the reference average and using a set confidence interval. The study first documents the significant spatial and temporal variability of the reference distribution spread and shape. It warrants the need for a finer assessment at the individual case level to characterise specific situations beyond the classical bulk

comparison using determinist "best" reference estimates. The probabilistic comparison allows for a more contrasted assessment than the deterministic one. Specifically, it reveals cases where the ARPEGE simulated values falling within the deterministic confidence range actually correspond to extreme departures in the reference distribution.

## 1 Introduction

Fundamental drivers of the climate system variability, such as atmospheric water cycle, are still not well understood. They are

associated with uncertainties that hampers climate predictions with consequences for society. An essential ingredient of the Earth's hydrological cycle, water vapor is the principal greenhouse gas and exerts a fundamental control on the distribution of temperature (Held and Soden, 2000; Pierrehumbert, 2011; Allan, 2012; Stevens and Bony, 2013). The radiative importance of the atmospheric water in maintaining the thermal energy balance of the Earth system is undebated. The connection between temperature, water vapor and infrared radiation creates a positive feedback that further warms the global climate from an

external forcing (Hartmann et al., 2013). In addition, cloud-moisture interactions and their associated processes are diverse





(Bony et al., 2015; Sherwood et al., 2010; Sherwood et al., 2020), and their representation in numerical models bear strong constraints on the local scales of weather forecasts and on global climate sensitivity (Stevens and Schwartz, 2012).

The accuracy of meteorological forecasts and climate projections relies on parametrization schemes or model physics. Assessing their accuracy is routinely performed by comparing the simulated geophysical fields to an observed reference derived from ground-based measurements or remote sensing techniques (Randall et al., 2007). When considering remote sensing techniques as reference, the comparison to numerical simulations may be performed in two way : (i) in the geophysical space, which means that the model geophysical variables are evaluated directly against remote sensing estimations based on a retrieval scheme, or (ii) in the observation (e.g., radiance) space, which means that a forward model is used to convert the simulated atmosphere into synthetic remote sensing measurements (Morcrette, 1991; Soden and Bretherton, 1994 ; Brogniez et al., 2005; Chepfer et al., 2008; Bodas-Salcedo et al., 2011; Jiang et al., 2012; Tian et al., 2013; Steiner et al., 2018).

On one hand, a retrieval scheme can be an inversion algorithm that relies on incomplete representations of the atmospheric variability (see for instance Stephens and Kummerow, 2007 for precipitation). On the other hand, the model-to-satellite approach relies on the accuracy of the forward model to simulate remote sensing observations for a given atmospheric state (Weng, 2007), while strong uncertainties may remain (Geer and Baordo, 2014; Brogniez et al., 2016).

In any case, the comparisons usually involve spatial and/or temporal averaging to adapt the resolutions of the two datasets (the reference and the model), sometimes including error bars. Moreover, common assessment practices typically use bulk comparison metrics (e.g. correlation, bias) to assess performances over a given spatial and temporal domain.

The present work focuses on atmospheric relative humidity (RH). There is an extensive body of literature on the use of relative humidity estimated by space-borne instruments to evaluate climate models (among others Soden and Bretherton, 1994; Brogniez et al., 2005; John and Soden, 2006; Jiang et al., 2012; Tian et al., 2013; Steiner et al., 2018…). However, the comparison generally provides limited insight in their error characteristics for several reasons:

1. First, an objective assessment requires an independent reference, which may not be verified when satellite remote sensing observations that are already incorporated in the model via an assimilation step are re-used to assess its accuracy.

2. Second, metrics such as correlation and bias are often applied without necessarily checking the relevance of such criteria. For example, the magnitude of the bias as an additive model-to-reference difference may be challenging to assess objectively at the primary satellite scale. The linear correlation is generally insufficient to describe the non-linear and heteroscedastic dependence structure between the model estimates and the reference.

3. Third, the model product is often assumed to be uniform and display homogeneous properties over the spatial and temporal domain of comparison. Bulk metrics such as correlation and bias are computed over samples that actually gather a variety of atmospheric situations (vertical structure, moisture, etc.) for which the model is likely to behave differently through its assumptions. Hence bulk error metrics lack specificity and depict averaged space/time properties while the errors tend to be non-stationary and sensitive to parameters not accounted for in the assessment



65        formulation. Therefore, the representativeness of any deterministic assessment of model RH is confined to the time and space domain over which it is performed, with limited extension over other regimes, regions, seasons, etc. (see Kirstetter et al., 2020 for a discussion on precipitation)

A probabilistic description of the reference RH is most appropriate to acknowledge the possible range of reference values.

This approach also explicitly accounts for deterministic uncertainties, making the diagnosis more documented and precise, ultimately contributing to the improvement of climate and weather forecasting models. This paper presents an assessment of the simulated RH using such probabilistic approach. The method is developed and tested to assess a sample of simulations of the global model ARPEGE ("Action de Recherche Petite Echelle Grande Echelle"), the numerical weather forecasting system developed by Météo-France (the French national weather service; Bouyssel et al., 2021). For this assessment, density functions

of reference RH are derived from the brightness temperatures measured by SAPHIR ("Sondeur Atmosphérique du Profil d'Humidité Intertropicale par Radiométrie"), the microwave sounder onboard the Megha-Tropiques satellite orbiting over the tropical belt (Roca et al., 2015).

This paper is divided into five sections. The datasets, and the matching procedure between SAPHIR probabilistic relative

humidity (RH) estimates and ARPEGE simulations are presented in section 2. The probabilistic method is introduced and confronted to the deterministic comparison in section 3. Section 4 discusses the results of the two comparison methods and the added-value of the probabilistic method. Concluding remarks are then drawn in section 5.

## 2 Data

ARPEGE 6-hourly instantaneous RH fields simulated at 6-hour lead time for the months April-May-June 2018 serve as a testbed for evaluating the numerical weather forecast model.

### 2.1 SAPHIR probabilistic RH estimates

SAPHIR is the microwave moisture sounder instrument onboard the Megha-Tropiques satellite, which observes the tropical (30°N to 30°S) atmosphere since October 2011 with a high revisit frequency. Megha-Tropiques is operated jointly by CNES

and ISRO (Desbois et al., 2007; Roca et al., 2015). SAPHIR measures across-track the upwelling radiation in the 183GHz water vapor absorption line over a 1700km swath. Each scanline is composed of 130 non-overlapping footprints whose nominal size is 10km at nadir and which deforms into an ellipse of 14.5x22.7km2 on the edges of the swath. SAPHIR samples with 6 channels ranging from 183.31 +/- 0.2 GHz to 183.31+/-11GHz. This original sampling allows a better vertical sounding of the tropical atmosphere compared to operational sounders like MHS and AMSU-B (3 channels) or ATMS (5 channels) (Brogniez

et al., 2013).





The measured brightness temperatures (BT) are translated into RH profiles for clear sky conditions as well as cloud-covered situations, as soon as cloud hydrometeors are small enough to not scatter the upwelling microwave radiation significantly. These conditions are associated with deep convection, with or without overshoots, and are detected from the BTs following Hong et al. (2005) and Greenwald and Christopher (2002). Therefore, RH profiles are estimated for every footprint of SAPHIR

if no deep convection is detected. The RH profiles are made of six relatively wide atmospheric layers ranging between 950 and 100hPa (100-200hPa; 250-350hPa; 400-600hPa; 650-700hPa; 750-800hPa; 850-950hPa) defined from an analysis of the channels' weighting functions (Sivira et al, 2015). The retrieval of RH profiles is based on a multivariate regression scheme that provides the parameters of a Beta probability density function (PDF) of the estimated RH for every footprint and pressure layer. This PDF allows to account for the spread and asymmetry of the atmospheric RH distribution that represent the

uncertainty of the retrieval scheme and the radiometric noise. The probabilistic profiles of RH have been compared to a large ensemble of radiosounding observations as well as to a sample of airborne lidar measurements to evaluate the retrieval (Brogniez et al., 2016; Stevens et al., 2017). The bulk biases lie in the range [0.3-10] %RH, with the bias depending on the pressure range and the vertical gradient of moisture.

## 2.2 ARPEGE simulated RH

The ARPEGE model is the operational global model operated by Météo-France since 1992 (Bouyssel et al., 2021). This model is characterized by a stretched and tilted horizontal grid and by a hybrid pressure terrain-following vertical coordinate system. The vertical grid is composed of 105 levels and the mesh of the horizontal grid has a 5 km resolution over Europe and a 24 km resolution elsewhere. Forecasts are initialized with a four-dimensional variational system (Courtier et al., 1992) with 6-hour windows and run up to a +102-hour forecast range. In the Tropics (30°N-30°S) and at this forecast range, the ARPEGE biases

(resp. rmse) on RH fields range between -5 and +5 % (resp. 5 and 25%) with respect to both radiosondes and the ECMWF analysis (Chambon et al., 2014).

For the purpose of this study, the 6-hourly forecasts of atmospheric RH have been projected on a regular horizontal 0.25° x 0.25° grid, and onto a regular vertical grid of 50 hPa resolution from 950hPa up to 100 hPa to match to the vertical resolution of the SAPHIR RH profiles. The vertical averaging implies that the results of the comparison are valid at the resolution of the

SAPHIR RH profiles.

## 2.3 Co-location

SAPHIR footprints are aggregated to match ARPEGE's 0.25 x 0.25° grid, as illustrated on Fig 1.a. A one-hour time window centered around each ARPEGE simulation is applied to SAPHIR pixels to avoid errors induced by shifts in moisture patterns due to global and local processes. The histogram in Fig 1.b shows the distribution of the number of SAPHIR footprints that

fall within ARPEGE gridboxes. This number varies from 0 to a maximum of 20, with a majority of sample sizes lying between 1 and 10 and the most frequent being 5.



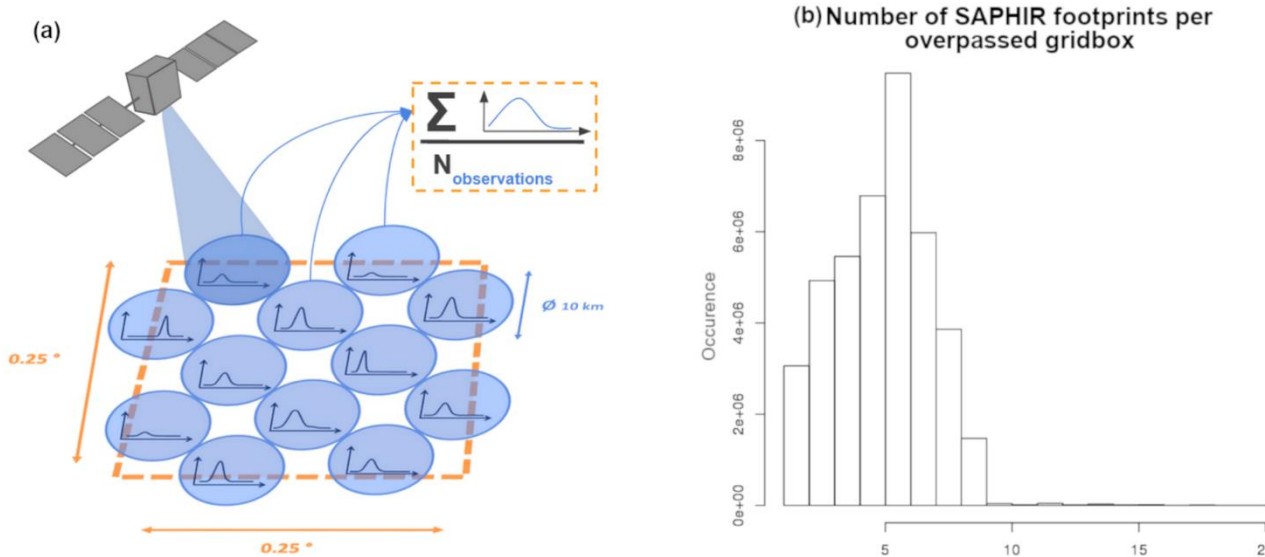

**Figure 1: (a) Colocalization diagram of SAPHIR's PDF and ARPEGE grid and (b) distribution of the sample size of SAPHIR footprints in ARPEGE gridboxes.**


Within each model gridbox, all the footprint RH PDFs are averaged together to compute an unconditional distribution of the mean RH at the ARPEGE scale. The averaged PDF encompasses all the available information of the reference RH such as the mean (first moment), uncertainty, and extremes.

## 3 Methodology

### 3.1 Statistical approach

### 3.1.1 Mathematical principles

The differences and complementarities between the deterministic and the probabilistic comparison approaches are illustrated in Figure 2. At any given pixel and timestep, the ARPEGE model value is noted $RH_{mod}$. The corresponding retrieval taken as reference $RH_{obs}$ is a random variable described by its cumulative distribution (CDF; defined in [0,1]) noted $F_{RH}$.


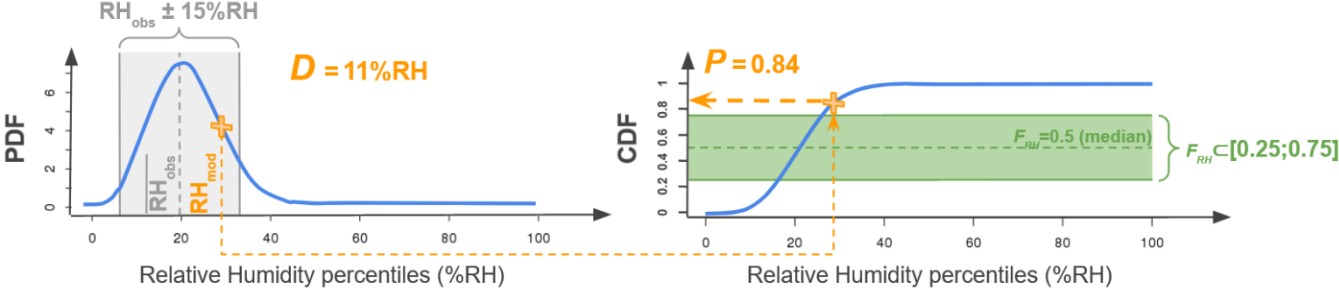

**Figure 2: Single-gridbox example (grid point situated 24.75 to 25 °N 114.5 to 114.75 °W and 400-600 hPa for the simulations of 2018/04/01 at 00 UTC) showing the projection of $RH_{mod}$ (orange cross) onto its associated reference distribution $RH_{obs}$ (blue curve) in terms of PDF (left panel) and CDF (noted $F_{RH}$; right panel). The reference expectation (first moment of the distribution) noted $\overline{RH_{obs}}$ is drawn as the grey dotted line centered within a ±15 %RH range (grey shade) on the PDF. The expected value (median) of the distribution is shown as the green dotted line on the CDF and the 0.25-0.75 quantile interval is shaded in green.**

The value $F_{RH}(RH_{mod})$ represents the probability $P$ that that $RH_{obs}$ takes on values lower or equal to the $RH_{mod}$:

$$F_{RH}(RH_{mod}) = \Pr(RH_{obs} \leq RH_{mod}) = P \tag{1}$$

The probability $P$ indicates the position of $RH_{mod}$ within the distribution of $RH_{obs}$. A probability value $P{\sim}0.5$ indicates that $RH_{mod}$ is close to the median that is a representative value of $RH_{obs}$. A probability value $P < 0.1$ (resp. $> 0.9$) indicates that ARPEGE probably underestimates (resp. overestimates) the reference $RH_{obs}$ as there are less than 10% chances that the $RH_{obs}$ takes on lower (resp. greater) values.

In order to compare the probabilistic method to a more classic approach, a simple deterministic comparison is used as a benchmark. The mean reference value $\overline{RH_{obs}}$ is calculated as the first moment of the PDF and it is taken as the reference in the deterministic comparison.

Eq. (2):

$$D = RH_{mod} - \overline{RH_{obs}} \tag{2}$$

The deterministic bias $D$ is defined as the difference between the $RH_{mod}$ and the mean observed value $\overline{RH_{obs}}$. A $D$ close to 0 indicates that the $RH_{mod}$ is close to the mean reference $\overline{RH_{obs}}$. An a priori deterministic ± 15 %RH confidence interval (grey shading on Fig. 2a) centered around $\overline{RH_{obs}}$ is chosen to account for uncertainties in the reference. It is a reasonable value that can be applied to the full RH profile. Note the retrieval uncertainty is lower in the mid-tropospheric layers compared to the edges of the profiles (Sivira et al., 2015). In terms of deterministic comparison $D < -15$ %RH (resp $> 15$ %RH) means that



$RH_{mod}$ significantly underestimates (resp. overestimates) the reference $\overline{RH_{obs}}$ average observation as it falls outside its confidence interval.

Compared to a deterministic comparison between ARPEGE's $RH_{mod}$ and the reference $\overline{RH_{obs}}$, $F_{RH}(RH_{mod})$ objectively quantifies the significance of the departure of ARPEGE w.r.t. the reference $RH_{obs}$ central value by factoring in (normalizing by) the spread of the distribution. It allows us to quantify the occurrence of extreme biases from the model while accounting for the tails of the distribution. By accounting for the complete reference distribution (and its characteristics like the spread and asymmetry), this probabilistic formulation allows comparison with greater resolution, sharpness and discrimination than a deterministic comparison at the pixel level.

### 3.1.2 Application to a single gridbox

The deterministic comparison and the CDF-based comparison are applied to each ARPEGE grid point. Figure 2 illustrates further the complementarity of the two approaches, for a representative case.

For any given ARPEGE gridbox the values $P$ and $D$ are computed from eq. (1) and eq. (2). As $RH_{mod}$ rarely falls on a $RH_{obs}$ percentile, $P$ is calculated with a linear interpolation between the two encompassing percentiles. A probabilistic confidence interval for the reference is defined at every ARPEGE gridbox, as the interquartile [0.25; 0.75] (green shading in Fig. 2b). This interval encompasses 50% of the $RH_{obs}$ distribution centered around the median ($F_{RH} = 0.5$, green dotted line).

In the example shown in Fig. 2, ARPEGE's $RH_{mod}$ is rather close to the mean reference value as it lies within 15 % RH from $\overline{RH}$ ($D = 11\%RH$, see Fig. 2a). However, the projection of $RH_{mod}$ onto the CDF (Fig. 2.b) indicates that it is located in the upper quartile of the distribution outside the [0.25; 0.75] reference confidence interval (Fig. 2b). $RH_{obs}$ has a low probability of $P=0.16$ of being lower than or equal to $RH_{mod}$. In other words, at this gridbox and time the probabilistic approach indicates that ARPEGE has a fairly high probability of overestimating RH while the deterministic comparison indicates an acceptable difference with the averaged value $\overline{RH}$. This example illustrates how the probabilistic comparison increases the information content in the reference by explicitly accounting for the reference uncertainty, which leads to a different conclusion than with the deterministic comparison that is based on a constant a priori uncertainty.

### 3.2 Applied methodology

### 3.2.1 Precipitation masking

As underlined in section 2.1, the retrieval of RH profiles from SAPHIR measurements is performed for both clear sky and cloud-covered areas, to the extent of scattering by large hydrometeors produced by convective activity is negligible (Greenwald



and Christopher, 2002; Hong et al., 2005). Therefore, all ARPEGE gridboxes associated with rainfall rates strictly above 0 mm/h are filtered out.

### 3.2.2 Temporal statistical accumulation

The comparison method is applied to a spatiotemporal domain covering the tropical belt over 3 months (April-May-June 2018). $D$ values are calculated at each gridbox and timestep and are averaged over time into a $\overline{D}$ value representing the average departure between ARPEGE's $RH_{mod}$ and the reference at this gridbox. In terms of the probabilistic comparison, $P$ values are also calculated at each gridbox and timestep and are aggregated over time to compute a PDF (Kirstetter et al. 2015), whose mode $P_{3M}$ (3M stands for "3 months") is kept. Figure 3 illustrates this process.


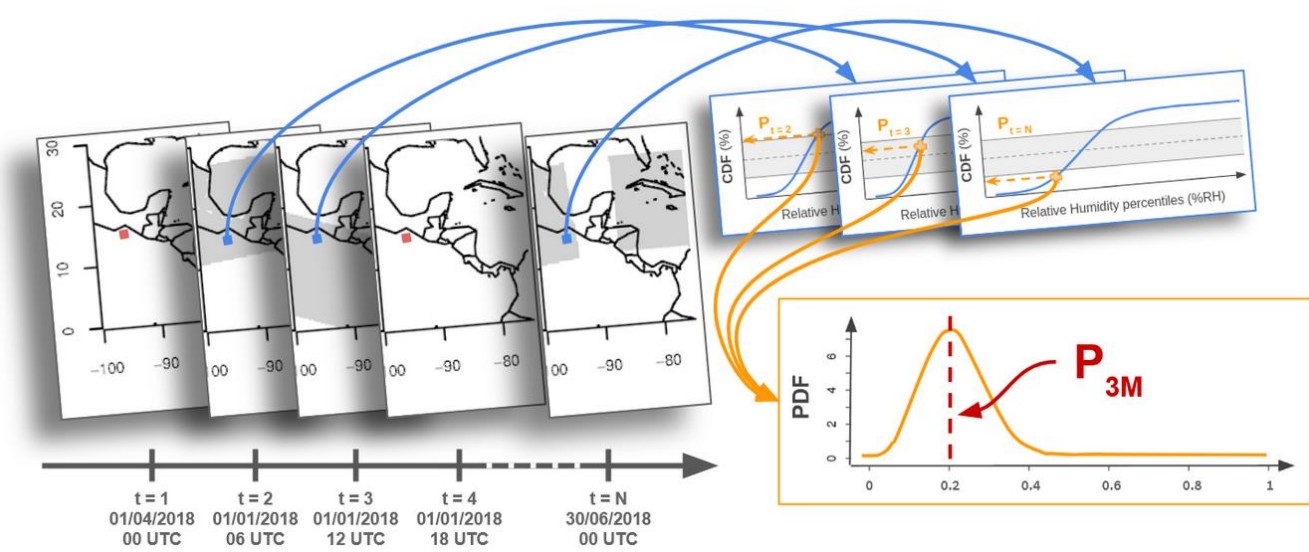

**Figure 3: Diagram presenting the spatial and temporal aggregation method for a single grid point (blue square when the gridbox was passed over by SAPHIR, red when not and/or filtered out). $P_{3M}$ is the mode of the distribution aggregated over all the considered time steps for this gridbox.**

**4 Results**

### 4.1 Uncertainty related to the assumption of gaussian distributions

In deterministic comparison settings, the uncertainty may be defined based on a priori assumptions, instrumental biases and retrieval errors. This uncertainty is often assumed to be multiplicative w.r.t to the reference value, and that the underlying density function is unimodal, symmetric, and follows a Gaussian model so that the uncertainty is defined as a standard

deviation. With the probabilistic approach, the uncertainty can be defined with the Inter-Quartile Range (*IQR*) of the





distribution calculated from the CDF at each time step and each gridbox (see section 3.1.2). Note that no assumption is made on the shape of the distribution in this case, hence the uncertainty is objectively and robustly quantified.

The smaller the *IQR*, the narrower the distribution and the smaller the uncertainty of the reference. Values of the IQR can be compared with the deterministic 15 %RH uncertainty. If the *IQR* is greater (resp. lower) than 15 %RH, then the reference
distribution is broader (resp. narrower) than assumed with a set 15 %RH error.

Figures 4.a and 4.b show an example of reference $\overline{RH_{obs}}$ and the associated *IQR* calculated at a given time step (2018/04/01 at 00 UTC) and for a given atmospheric layer (650-700 hPa). Figure 4.c shows the frequency of occurrence for cases with *IQR* > 15 %RH at each gridbox over the three-month period April-June 2018.


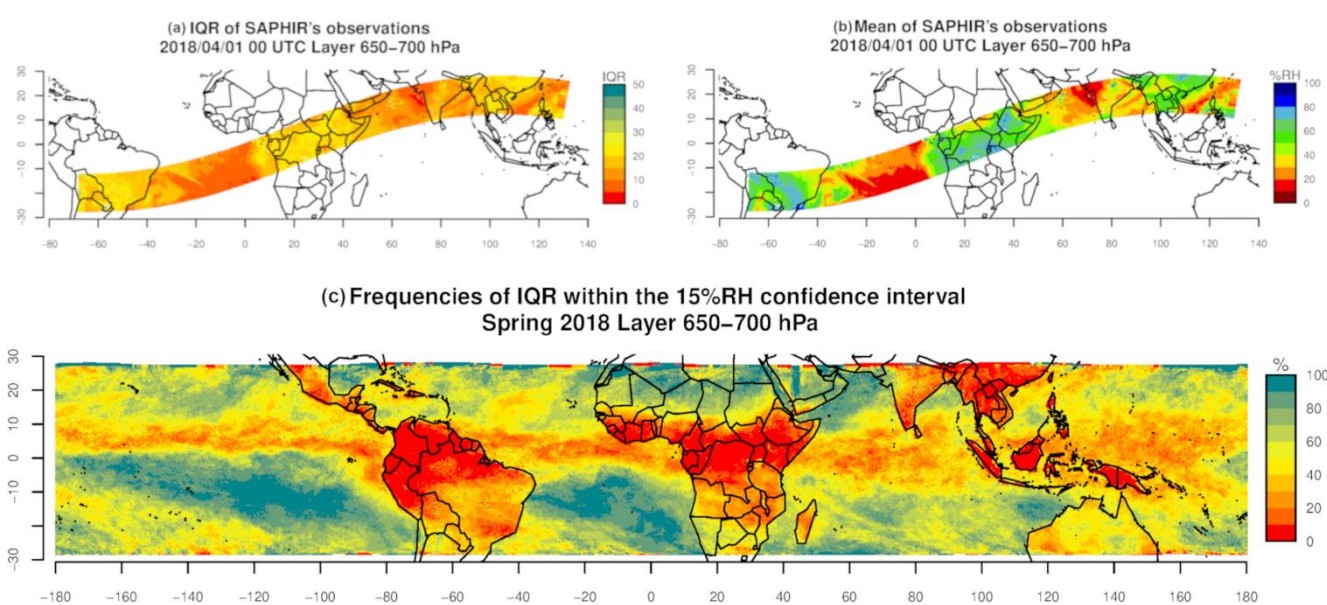

**Figure 4: (a) Spatial distribution of the inter-quartile range (*IQR*) and (b) mean of reference ($\overline{RH_{obs}}$) representing the observed spatial field of RH from SAPHIR on 01/04/2018 at 00UTC and at the 650-700 hPa layer (c) Frequency of *IQR* ≤ 15 %RH over the three-month period (April to June 2018)**


The spatial distribution of the *IQR* (Fig 4.a) shows contrasted areas that match the patterns of $\overline{RH_{obs}}$ (Fig 4.b). Low *IQR* values (*IQR* ≤ 15 %RH, appearing in red and orange on Fig 4.a) are consistently found where the atmosphere is dry, for example above the Atlantic Ocean and the Arabian sea. High *IQR* values (*IQR* > 15 %RH, in yellow to green on Fig 4.a) are found over moister atmosphere, especially above South America, Africa and South Asia. It suggests that while scaling relation between
RH and its uncertainty is relevant, a typical 15 %RH assumption lacks dynamics to represent the true uncertainty. The contrasted differences between the *IQR* < 15 %RH and *IQR* > 15 %RH areas show that the width of the $RH_{obs}$ distributions



vary significantly and are linked to the same processes. In addition, in the areas with $IQR > 15$ %RH the wider range of $RH_{obs}$ illustrates the equally wide diversity of underlying situations. These results illustrate the need to use a comparison method that adapts to the range of variability at each gridbox. Especially in the areas where the $IQR$ is high, using a deterministic approach

solely based on the distribution expectation does not capture the variety and complexity of situations and is thus ill-advised. However, we note that high $IQR$ values found in moister situations may also result from multiple causes such as the aggregation of distinct sub grid processes or higher retrieval errors (Sivira et al., 2015; Brogniez et al., 2016). Nonetheless, using a probabilistic approach make any comparison more specific to each situation by avoiding the use of a consistent simplification (i.e. a set confidence interval) that does not fit the underlying complexity.


The comparison of the two uncertainty approaches over the whole period (Fig. 4.c) confirms the spatial correlation between the $IQR$ and the classical patterns of the humidity field. This is particularly visible around the South Pacific and South Atlantic Highs where the proportion of $IQR$ under the 15 %RH threshold reaches 70% and even 100%. In these subsiding areas, the atmospheric RH is ruled by large-scale processes that have little to no instantaneous variability at the scale of our grid. It results

in more homogeneous conditions within the same gridbox that explains the narrower distributions of the retrievals (i.e. smaller $IQR$). The Intertropical Convergence Zone (ITCZ) appears through areas of low proportion (0 to 30% of the retrievals' dataset) of under-15 %RH $IQR$. High dynamics characterizing this zone result in smaller-scale processes that impact the RH field and result in heterogeneous conditions within the same gridbox and larger $IQR$. Most importantly, $IQR$ varies across space and time, and it can be partly linked to the RH field and explained by large and fine scale processes. These highlight the need for

a comparison method that exploits and takes into consideration the variability of the dataset and adapts the comparison to each situation.

One can note that while these results vary significantly depending on the atmospheric layer, they are coherent with the expected RH field patterns. For example, in the upper two atmospheric layers (100-200 hPa and 250-350 hPa), the homogeneous dryer

conditions result in almost all retrievals distributions having $IQR$ under 15 %RH. The two lower layers (750-800 hPa and 850-950 hPa), closer to the ground show strong ocean-continent contrasts. This contrast shows the difference of processes that depend on the surface, with extremely low frequencies of $IQR$ under 15 %RH above the continents. It suggests that the lower the layer, the wider the distribution of retrieved RH.

In all cases where the $IQR$ is above 15 %RH, flattened and possibly non-gaussian RH distributions may result in non-representative $\overline{RH_{obs}}$ values associated with varying confidence intervals that make the deterministic comparison less appropriate. These results show that these situations can be quite frequent and geographically distributed. The probabilistic method allows us to adapt the confidence to each distribution regardless of the width and shape, and thus improves the assessment accuracy both overall and at the gridbox/timestep scale.



## 4.2 Application to a single time step

Figure 5 shows the comparison results between the reference $RH_{obs}$ and ARPEGE's $RH_{mod}$ obtained with both deterministic and probabilistic methods.

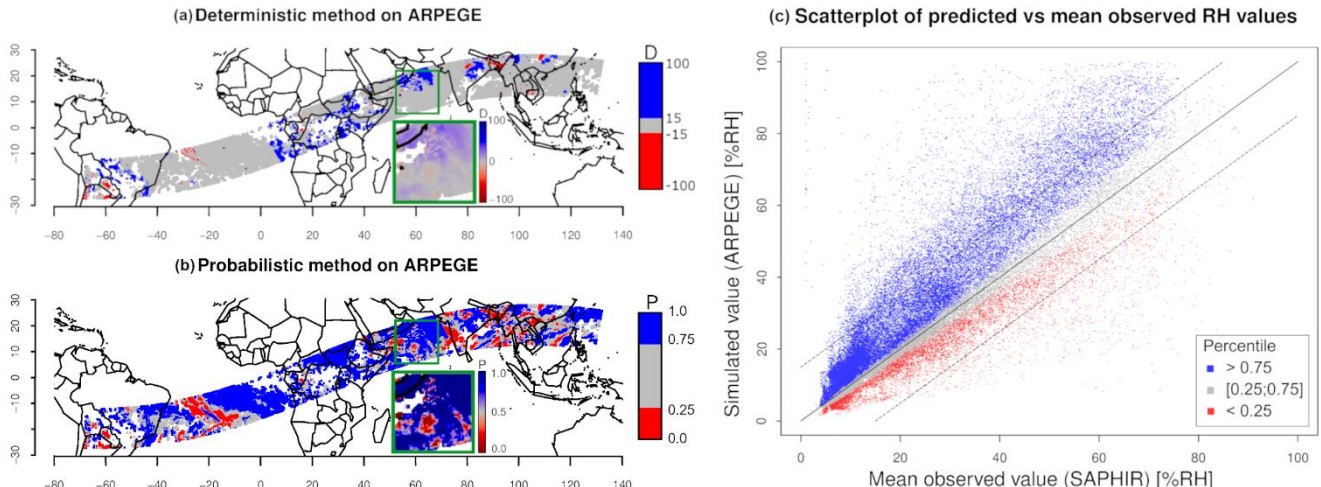

**Figure 5: Comparisons between SAPHIR's $\overline{RH_{obs}}$ and ARPEGE's $RH_{mod}$ on 01/04/2018 at 00UTC and for the 400-600 hPa atmospheric layer. (a) map of the difference $D$, (b) map of the probability $P$ and (c) scatterplot between $\overline{RH_{obs}}$ (x-axis) and $RH_{mod}$ (y-axis); the percentile associated to each point as $P > 0.75$ (blue), $0.25 < P < 0.75$ (grey), and $P < 0.25$ (red); the solid line represents the y=x line and the dotted lines delimit the $\bar{x} \pm 15$ %RH around values**

The comparisons are performed using wide discrete color bars in order to discuss the complementarity of the methods and not specific issues of the model. The deterministic approach shows that a majority of $RH_{mod}$ values are within the interval of $\pm 15$ %RH of $\overline{RH_{obs}}$ (grey areas on Fig 5.a). By contrast, the probabilistic approach reveals that only a few of the corresponding $P$ fall within the [0.25;0.75] probabilistic interval (blue or red on Fig 5.b). Large areas of deterministic reasonable differences such as in the Southern Atlantic are associated with $P > 0.75$, meaning that the corresponding $RH_{mod}$ has a high probability overestimating the reference. The inset in the maps is a zoom that highlights similar bias patterns with both methods but with higher contrasts with the probabilistic one. The patterns only appear in the deterministic results when using a continuous color scale that allows to show small differences w.r.t. the mean $\overline{RH_{obs}}$. These patterns can be observed with both methods in general, but the small differences between the $RH_{mod}$ and $\overline{RH_{obs}}$ prevent a robust diagnosis. The probabilistic results are particularly contrasted in comparison because these $RH_{mod}$ values fall into in the extreme quartiles ($P < 0.25$ or $P > 0.75$). The scatterplot (Fig. 5.c) further illustrates the added value of the probabilistic approach. The confidence interval set by $P$ within [0.25,0.75] (grey triangular shape) follows closely the $RH_{mod} = \overline{RH_{obs}}$ line, showing the overall consistency of the two methods. For small values of RH, this interval is much tighter around the line than at higher values. The confidence interval widens for higher values of RH, confirming that the probabilistic method is more specific than a deterministic approach with constant error bars





around the $RH_{mod} = \overline{RH_{obs}}$ line. Furthermore, situations where $P$ falls within the [0.25; 0.75] interval are less common when compared to the ± 15 %RH range defined around the mean. The blue and red dots that appear within the ± 15 %RH range indicate cases where $RH_{mod}$ are close to $\overline{RH_{obs}}$ ($D$ within [-15;15]) but fall in extreme $RH_{obs}$ distribution quartiles ($P$ outside

[0.25;0.75]). Most blue and red points (respectively above and under the ± 15 %RH range) show that a strong deviation from mean nearly always matches with extremes of the distribution ($P < 0.25$ or $P > 0.75$).

In short, the probabilistic method is consistent with the deterministic comparison on extreme biases and adds more information for cases where $RH_{mod}$ seems close to $\overline{RH_{obs}}$ .

**4.3 Comparison results for an extended period of time**

**4.3.1 Distributions**

The two methods are applied on the entire data set over the period from April to June 2018. The distributions of each method's results are represented as histogram (Fig 6.a) and rank histogram/Talagrand diagram (Hamill, 2001; Wilks, 2011; Kirstetter et al. 2015; Fig 6.b). Note that while the Talagrand diagram is often used to assess ensemble forecasts by comparing a single

reference to a distribution of forecast ensembles, its interpretation is similar when comparing a single simulated value to a reference distribution. This graphical method illustrates where the ARPEGE's $RH_{mod}$ falls in the distribution function $CDF$ of $RH_{obs}$. In case of perfect forecasts, each quantile represents an equally likely scenario for the ARPEGE model. The perfect case is a flat rank histogram indicating that the $RH_{obs}$ probability distribution is well represented by $RH_{mod}$.

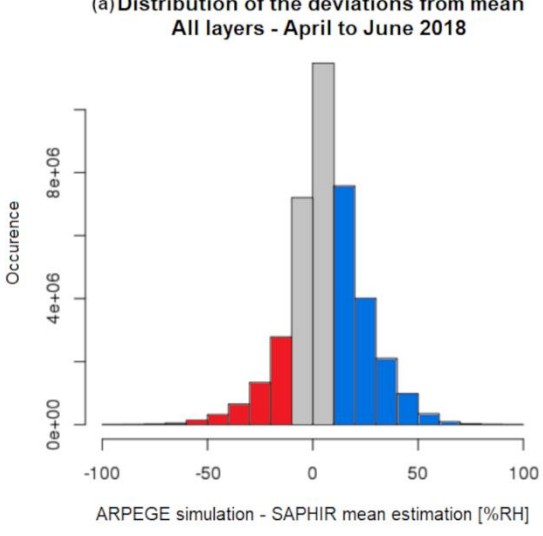

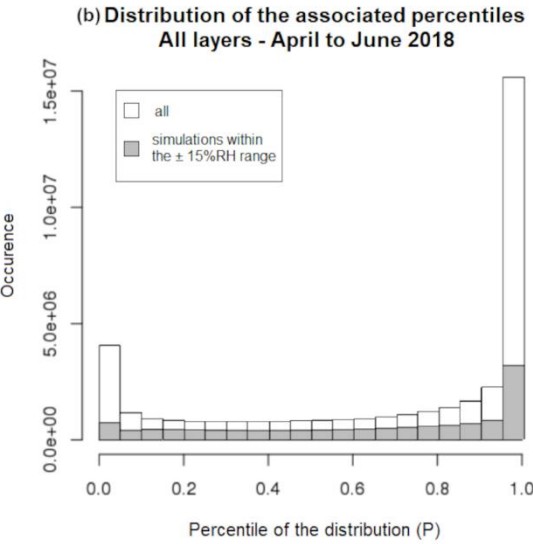


Figure 6 : Distribution of the results from each comparison method applied on all layers over the period April to June 2018 : (a) histogram of the deviations $D$ from the mean (in red : $D < $ -15 %RH ; grey : -15 %RH $\leq D \leq$ 15 %RH ; blue : $D > 15$





%RH) and (b) rank histogram/Talagrand diagram of $P$ (in grey: values of $P$ when $D$ within [-15;15] %RH; in white: for all values of $D$).

As seen on Fig 6.a, the deterministic difference $D = RH_{mod} - \overline{RH_{obs}}$ displays a unimodal and symmetrical distribution centered around 0. All layers combined, more than 67%, of the $RH_{mod}$ deviate by less than 15 %RH from the mean estimate $\overline{RH_{obs}}$. Nearly 20% of all $RH_{mod}$ are too high by more than 15 %RH and less than 2% by more than 50 %RH. Yet the Talagrand diagram on Fig 6.b shows that the associated percentiles $P$ display a bimodal distribution with maximum frequencies concentrated at the two extreme ends. Half the ARPEGE's $RH_{mod}$ are associated with extreme percentiles of the $RH_{obs}$

distribution ($P<0.05$ or $P>0.95$). Less than a quarter (23.3%) $RH_{mod}$ fall within the centered half of the $RH_{obs}$ distribution. The Talagrand diagram allows us to draw direct conclusions on the reliability of the forecasts. Assuming that the forecasts are spread far enough in both time and space to be considered independent from each other, the probabilities of finding each percentile of the histogram should be fairly equal giving the diagram a flattened aspect, which is not observed here. The U-shape indicates that the extreme classes are over-represented and compensate for the under-estimations of the central quantiles

(Candille & Talagrand, 2005). There is also a tendency of ARPEGE to overestimate RH. These features remain when focusing on $RH_{mod}$ values within the 15 %RH from the mean estimate (grey histogram on Fig. 6.b). It confirms that the overestimation is present within the deterministic confidence range. Validating $RH_{mod}$ values by their proximity to the mean reference (-15 < $D$ < +15 %RH) is not sufficient to assess their accuracy, especially when they fall in the extreme percentiles of the associated distribution ($P < 0.25$ or $P > 0.75$).


The comparisons are performed independently for each atmospheric layer in Fig. 7. The distributions are represented as boxplots in Fig. 7, with the width of the box is defined by the first and third quartiles and the length of the whiskers by either 1.5 x $IQR$ or the most extreme value if under. It highlights a variability in terms of both shift and spread.

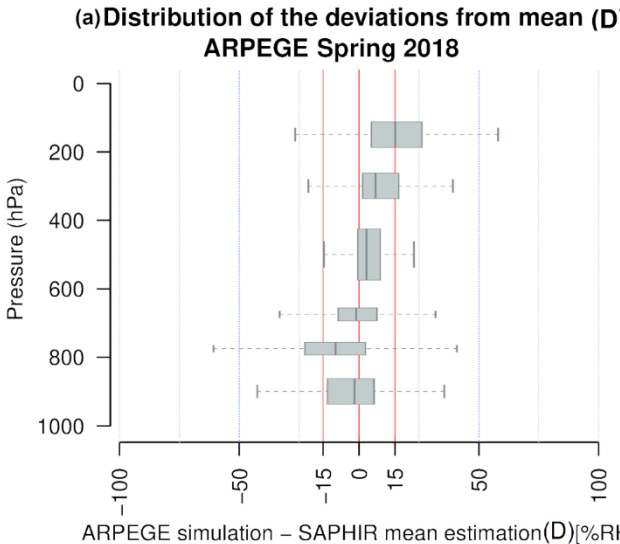

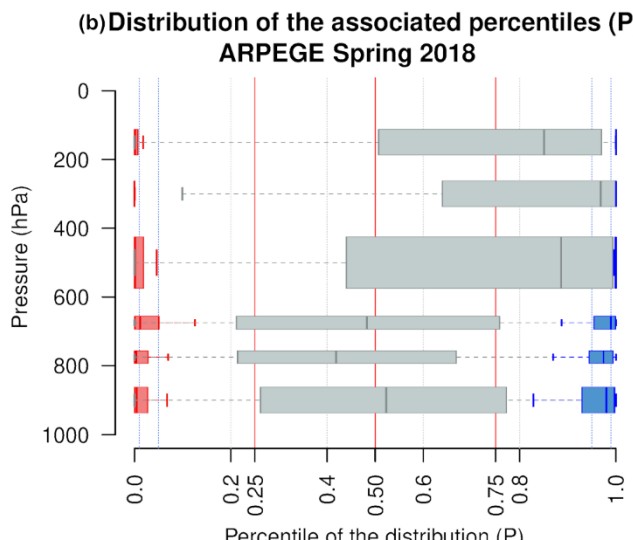






**Figure 7: Distribution the results of each method of comparison applied on the period April to June 2018 represented as boxplots for each layer: (a) deviations $D$ from mean and (b) associated percentiles $P$ divided into three distribution types depending on the category of $D$ (red: $D < $ -15 %RH; grey: -15 %RH $\leq D \leq$ 15 %RH; blue: $D > $ 15 %RH). The weights of each category within their layer's distribution are indicated in Table 1**

The distributions of the deviations from the mean $D$ (Fig. 7.a) show a tendency of the model to overestimate $\overline{RH_{obs}}$ in the upper tropospheric layers (100-200 hPa, 250-350 hPa), to close in on $\overline{RH_{obs}}$ in the mid-tropospheric layers (400-600 hPa) and to slightly underestimate $\overline{RH_{obs}}$ in the lower tropospheric layers (650-950 hPa). The highest layer (100-200 hPa) has the most off-centered results, with half (50.3%) of the simulated values overestimating $\overline{RH_{obs}}$ by more than 15 %RH. The layer 750-800 hPa has more than a third (38.7%) of the simulations more than 15 %RH below $\overline{RH_{obs}}$ ($D < $ -15 %RH). The other layers

have the majority of their distribution well within the ±15 %RH range.

The distributions of the associated percentiles are wider (Fig. 7.b) and offer a deeper understanding. The comparison results are divided into three categories that follow the deviation from the mean intervals. They are drawn separately to highlight the consistency of the two methods' extreme results. The ARPEGE's $RH_{mod}$ distant from $\overline{RH_{obs}}$ by more than 15 %RH ($D$ outside

[-15%;15%]) almost always fall in either one or the other extreme end of the estimated distribution (red and blue boxplots in Fig. 7b). Table 1 provides, for each atmospheric layer, the portion of the distributions that falls inside or outside the interval [-15%;15%]. The proportion of extremes vary from one layer to the another. It allows us to separate the two maxima of the U-shape of the rank histogram/Talagrand diagram (Fig. 6.b): the left-hand extreme is largely influenced by the under-estimated values in the lower layers (750-800 hPa, 850-950 hPa) while the right-hand extreme is driven by the over-representation of

over-estimations in the higher layers (100-200 hPa, 250-350 hPa).

|  | $D < $ -15 %RH | -15 $\leq D \leq$ 15 %RH | $D > $ 15 %RH |
|---|---|---|---|
| **100 - 200 hPa** | 1.32% | 48.34% | 50.34% |
| **250 - 350 hPa** | 0.41% | 71.62% | 27.97% |
| **400 - 600 hPa** | 0.87% | 87.66% | 11.47% |
| **650 - 700 hPa** | 13.23% | 74.31% | 12.46% |
| **750 - 800 hPa** | 38.67% | 52.37% | 8.96% |
| **850 - 950 hPa** | 22.34% | 70.45% | 7.21% |

**Table 1: Part (in %) of the distributions within the three categories of differences D for each atmospheric layer.**

An added value of the probabilistic approach resides in the contrast and variability of the results within the ±15 %RH range.

As previously shown, the implicit hypothesis in a deterministic method does not allow to narrow the confidence interval further than 15 %RH, thus preventing a more contrasted diagnosis of the model's outputs. The grey boxplots highlight the great variability in cases contained within the deterministic interval, that is not revealed by a small difference $D$ to the mean estimate





$\overline{RH_{obs}}$. Even though the distributions of $P$ are fairly spread, in the highest layers (above 400 hPa) they have a strong tendency to fall on higher percentiles of the reference distributions: more than 75% of the simulations of the first two layers (between

100 and 350hPa) are above 0.75 and 65% into the upper 0.05. It confirms the general overestimation displayed by the deterministic results in these layers. The underestimations of the lower layers are also clear, with 54.4% and 39.1% of the simulations falling into the lower quarter of the reference distribution in the second to lowest layer (750-800 hPa) and lowest layer (850-950 hPa), respectively. The distribution of the 650-700hPa layer is the most centered, yet is spread and still has more than half of the associated percentiles (66.2%) outside the 0.25-0.75 interval.


The 400-600hPa layer has the narrowest distribution of deviations from the mean within the ±15 %RH range. However, the distribution of the associated percentiles is mostly located towards the higher half (64.7% above 0.75) and nearly half of the simulations (47.0%) fall onto the upper extreme 0.05%.

### 4.3.2 Maps: Deterministic results

The two maps in Fig. 8 show the results of the deterministic approach in terms of the average deviation from $\overline{RH_{obs}}$ ($\overline{D}$) (Fig. 8.a) and frequency of $RH_{mod}$ within the ± 15 %RH range (Fig. 8.b) for layer (400-600 hPa; other layers can be found as supplementary material). The combined results of the three-month period show recognizable patterns.

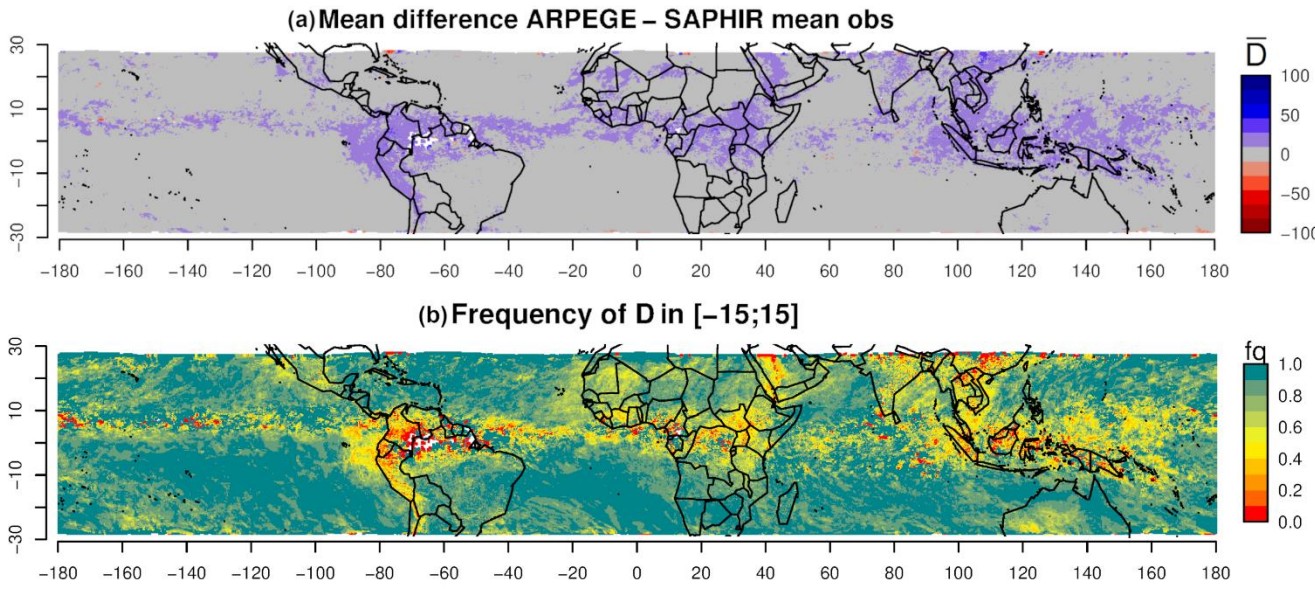

**Figure 8: Deterministic comparisons for layer 400-600 hPa over the April to June 2018 period. (a) Map of the mean difference** $\overline{D} =$

$\overline{(RH_{mod} - \overline{RH_{obs}})}$ **and (b) frequency of instantaneous difference** $D$ **falling within [-15;15] over the 3-month period.**





The majority of the average deviations $\bar{D}$ are close to zero (displayed in grey in Fig. 8.a). These regions have 80% to 100% of single-time-step $D$ within the ± 15 %RH range (light to dark green in Fig. 8.b). On average, the model's overestimated RH values (blue) are localized above known convective areas (ITCZ, South Pacific Convergence zone, or due to south-American coastal orographic convection processes). These on-average overestimated zones are characterized by low frequencies of single-time-step $D$ within the ± 15 %RH range (50% and less, appearing as yellow and orange in Fig. 8.b). It indicates that this potential bias occurs at least half the time over the study period. Only a small number of pixels show an on-average underestimation (red), as predicted by the distribution of the deviation results (see Fig. 7.a and the discussion in section 4.3.1). The results of this method reveal a slight moist bias in the convective zones, but mostly validate ARPEGE simulations everywhere else outside these areas.

### 4.3.3 Maps: probabilistic results

Figure 9 represents maps of the probabilistic comparison method in terms of spatial distribution of the mode $P_{3M}$ of the PDF calculated from the single-time-step P aggregated over the 3 months, and the frequencies of $P$ within the three categories ($P > 0.25$, $0.25 \leq P \leq 0.75$, $P > 0.75$).



**Figure 9: Maps of the probabilistic method for the layer 400-600 hPa applied over the April to June 2018 period. (a) Mode $P_{3M}$ and frequencies of single-time-step $P$ within (b) the middle-half ($0.25 \leq P \leq 0.75$), (c) lower quarter ($P < 0.25$) and (d) upper quarter ($P > 0.75$) of the distribution of estimates**




The probabilistic comparison method highlights a majority of contrasted extreme values, which indicates a high probability of ARPEGE's $RH_{mod}$ falling onto one or the other extreme quarter of the $RH_{obs}$ CDF. The maps show organized and contrasted spatial patterns with a majority of overestimated areas (blue to dark blue in Fig. 8.a) along with some under-estimated patches (red to dark red). The overestimations are predominant with a frequency of at least 40% in most areas (Fig. 8.d). The convective

zones are overestimated with the deterministic method but seem more complex with probability P falling on both the middle half and the upper extreme segments of the $RH_{obs}$ CDF.

The red patch located south of the African continent (Fig. 9.a) indicates a recurring underestimation of the model, with $RH_{mod}$ values falling in the lower quarter of the $RH_{obs}$ CDF. These underestimations happen at least half of the time during the studied period (in bright red in Fig. 9.c). The frequencies of simulated values within the middle half part of the reference distribution

are fairly low (mostly under 0.4, Fig. 9.b). The frequencies of P > 0.75 are almost null (Fig. 9.d). Other underestimated areas can be observed, e.g. above the Caribbean and above the Pacific Ocean, east of Australia. Above the Indian ocean, even though most $D$ are within the ± 15 % RH range, the situation appears to be significantly more contrasted with the probabilistic approach. Again, the deterministic approach only finds close $RH_{mod}$ values to $\overline{RH_{obs}}$, the probabilistic method assigns high probabilities for $RH_{mod}$ to fall into one or the other extreme of the $RH_{obs}$ distribution. In this area, both high frequencies of P in the lower

and the upper quarters of the distribution are found spatially close to each other and without any particular pattern (speckled aspect in the Indian Ocean, Fig. 9.a, 9.c and 9.d).

These various problematic areas do not particularly stand out when solely using a deterministic comparison approach. The probabilistic method allows for a more contrasted and detailed assessment. Note that the analysis of the results with regard to the model specificities, such as its parameterization of convection, are outside the scope of this paper.


## 5 Conclusion

This paper showcases the importance of considering all the reference information content through a probabilistic approach that considers the reference distribution to assess ARPEGE model simulations. The probabilistic reference is derived from finer-scale RH estimates aggregated into a probability density function at the ARPEGE spatial resolution. In widely used

deterministic comparison approaches, the reference distribution is only considered through its first moment (and sometimes its second moment) only. The improved assessment with the probabilistic approach is demonstrated by comparing the insights obtained on ARPEGE with that from a deterministic method involving the difference $RH_{mod} - \overline{RH_{obs}}$ and a ±15 % RH confidence interval.

First results highlight the inherent inaccuracy of solely using averaged references due to the important variability of spread and shape of the reference estimates. By computing the inter-quartile range ($IQR$) for the whole reference dataset, it was found that the spread of the PDFs varies significantly and are linked to the RH magnitude, with wider distributions in moist areas




and narrower in drier conditions. A deterministically set confidence interval is relevant to the variability of the spread to some extent only. It promotes a comparison method that quantifies more precisely the deviation of the simulated value irrespective
of the reference distribution variability, spread and shape.

Both deterministic and probabilistic methods were confronted on a single time step and over the 3-month period. Most RH values simulated by ARPEGE fit within the ± 15 %RH confidence interval with a slight moist bias detected in the ITCZ. However, the probabilistic method reveals that $RH_{mod}$ values that differ from $\overline{RH_{obs}}$ by more than 15 %RH ($D < $ -15 %RH or
$D > $ +15 %RH) often correspond to the extreme 5% of the reference distributions ($P < 0.5$ or $P > 0.95$). The probabilities associated with $RH_{mod}$ values within the deterministic confidence range are often outside the probabilistic confidence interval [0.25 – 0.75], which highlights model biases. The highest layers (100-600 hPa) show high occurrence of probabilities within the upper quartile of the reference distributions ($P > 0.75$), allowing to conclude that ARPEGE overestimates RH in these layers. The middle and lower layers (600-950 hPa) have $P$ distributions more centered around the reference median but are
wider than the 0.25-0.75 interval. For these layers, the spatial distribution of the probabilistic results shows a likely overestimation of ARPEGE in convective areas and a tendency to underestimate specific subsiding systems. This last observation is not detected with the deterministic method and it adds new perspectives on potential biases of ARPEGE.

Overall, the probabilistic comparison allows a more contrasted and complete assessment. The bias structures that are revealed
fit known humidity patterns. A more complete analysis with regard to the model's specificities could help highlight areas of improvement. The method presented here can be generalized to different models, variables, and observations.

**7 Acknowledgements**

We thank CNES for its financial support through the Megha-Tropiques project and the national Aeris data center that hosts
the satellite data. Christophe Dufour (LATMOS/IPSL) contributed to the data processing. Computing resources of ESPRI/IPSL were greatly appreciated. SAPHIR data are available through the Aeris/ICARE ground segment of Megha-Tropiques (https://www.icare.univ-lille.fr/product-documentation/?product=SAPHIR-L2B-RH). Pierre Kirstetter acknowledges support from the NASA Global Precipitation Measurement Ground Validation program under Grant NNX16AL23G.



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
