# Peer review of "Novel assessment of numerical forecasting model relative humidity with satellite probabilistic estimates"

_Atmospheric Chemistry and Physics, 2021_

## Author Comment (AC1)

We would like to thank the anonymous reviewers for their very detailed comments on our work. The two reviewers have highlighted important imprecisions in the paper. We hope that our answers and the modifications added to the manuscript will help future readers better understand our work and avoid any misinterpretation. The updates appear in track-change mode in the article file.

This document presents our answers to RC2. For added clarity, we colour coded our answers as follows:

***Black: original comments from Anonymous Reviewer #2***
Blue: our direct answer
Green: corrections and/or additions as they appear in the updated manuscript. The new line number also appear.

**1. General Comments**

*This paper presents a novel method for assessing humidity fields from numerical weather prediction models with estimates from the SAPHIR instrument. The probabilistic methodology used to estimate relative humidity from SAPHIR is exploited to provide a new approach for model assessment. The methodology also allows for a confidence interval to be placed on comparisons where classical 'bulk' comparisons.*

*This study demonstrates an innovation that yields more nuanced results for satellite and model inter-comparisons. This is important for relative humidity, where uncertainties in satellite measurements can be as high as 10% RH for some instruments (especially heritage infrared sounders). Overall, I find that this study is of scientific value and recommend it for publication, after all the issues that I have highlighted are addressed.*

**2. Specific Comments**

*Line 21: The final sentence in your abstract is illustrating a key point of your study but it is missing the "why" of its importance. Adding another sentence or editing this final one will make it more impactful.*

Thank you for the suggestion. Following this remark, we worked on the abstract and developed as follows:

**Lines 21 to 24:** "Specifically, it reveals cases where the ARPEGE simulated values falling within the deterministic confidence range actually correspond to extreme departures in the reference distribution, highlighting the shortcomings of the too-common Gaussian assumption on the reference error, on which most current deterministic comparison methods are based."

*Line 42: Why use a reference for precipitation when talking about humidity? There are plenty of water vapour retrieval algorithm papers that perform an inversion between an atmospheric stat vector and observed radiances/brightness temperatures. Please update.*

We understand and agree to that comment. The following references related to water vapour were added instead:

**References:**
Rosenkranz P. 2001. Retrieval of temperature andmoisture profiles from AMSU-A and AMSU-B measurements. IEEE Trans. Geosci. Remote Sens. 39: 2429–2435.

Lerner, J. A., Weisz, E., and Kirchengast, G., Temperature and humidity retrieval from simulated Infrared Atmospheric Sounding Interferometer (IASI) measurements, *J. Geophys. Res.*, 107( D14), doi:10.1029/2001JD900254, 2002.

Karbou F, Aires F, Prigent C, Eymard L. 2005. Potential of Advanced Microwaves Sounding Unit-A (AMSU-A) and AMSU-B measurements for atmospheric temperature and humidity profiling over land. J. Geophys. Res. 110: D07109, DOI: 10.1029/2004JD005318.

Divakarla, M., et al. (2014), The CrIMSS EDR Algorithm: Characterization, Optimization, and Validation, *J. Geophys. Res. Atmos.*, 119, 4953– 4977, doi:10.1002/2013JD020438.

**Lines 45-47: Averaging is not the only method used to get data on the same resolution. The discussion here does not include the use of averaging kernels, which are used to smooth model or in situ profiles relative to the vertical resolution of the satellite measurement. See "Rodgers, C.D. and Connor, B.J., 2003. Intercomparison of remote sounding instruments. Journal of Geophysical Research: Atmospheres, 108(D3)."**

We reworked the sentence to mention the use of averaging kernels. It now reads as follows:

**Lines 47 to 49:** "In any case, the comparisons usually involve spatial and/or temporal averaging, sometimes involving error bars or the use of averaging kernels to smooth models or in situ profiles relative to the vertical resolution of the satellite measurement (Rodgers and Connor, 2003)."

**Lines 65-67: It reads a bit strange when you talk about RH and then reference a precipitation paper for further discussion. If this is the only suitable reference there needs to be slightly more elaboration as to why. For instance, is the discussion point in the paper about representativeness but in the context of precipitation?**

Although we understand the concern, this a point for RH that is also valid for any geophysical variables. We developed the sentence that now reads:

**Lines 69 to 71:** "These issues are not confined to the study of RH but are, to an extent, common to those of all geophysical variables (see for instance Kirstetter et al., 2020 for a discussion on precipitation)."

**Line 93: A figure here might illustrate this point better for the channels on SAPHIR. Not all readers may be familiar with MW remote sensing, especially the 183 GHz region where the +/- values relate to where on the wings of the 183 GHz feature SAPHIR is sampling. Alternatively, the sentence could be updated to reflect this point and why it is done.**

Thank you for this suggestion. We think that adding a figure may not be necessary for the presentation of the instrument. We added precision to the following sentence, which now reads:

**Line 96 to 98:** "SAPHIR spectrally samples the 183 GHz line with 6 channels ranging from 183.31 +/- 0.2 GHz (close to the center of the line for upper tropospheric sounding) to 183.31 +/-11GHz (wings of the line for a deeper sounding)"

**Lines 96-108: Is the SAPHIR measurement noise (measurement uncertainty) used at all in the RH retrieval?**

Yes, the in-flight radiometric noises were used in the training phase of the model, as discussed in Sivira et al, 2015 and Brogniez et al, 2016.

**Line 115: "RH fields range between -5 and +5 % (resp. 5 and 25%)" what do the values in brackets relate to?**

The values in brackets relate to the RMSE, we however reorganized the sentence to avoid any misunderstanding:

**Line 126 to 128:** "In the Tropics (30°N-30°S) and at this forecast range, the ARPEGE biases on RH fields range between -5 and +5 % and rmse between 5 and 25% with respect to both radiosondes and the ECMWF analysis (Chambon et al., 2014)."

*Line 120: Does the vertical averaging account for SAPHIR weighting functions? – in a similar way to which upper tropospheric humidity is calculated?*

No. As detailed in Sivira et al (2015), the weighting functions of SAPHIR have only been used during the design phase of the RH retrieval scheme, to determine the 6 atmospheric layers from the set of 6 measurements (6 BTs for every footprint). A UTH product is also available from the 3 upper channels of SAPHIR (see Brogniez et al, 2015, JAMC, DOI: 10.1175/JAMC-D-14-0096.1), but we are analyzing here the RH profiles defined on fixed atmospheric layers. Therefore, over the former lines 120-121 we match two profiles of RH: one with only 6 wide layers (RH from SAPHIR) and the other one with 18 thin layers (RH from ARPEGE). The vertical averaging in thus only performed to make ARPEGE match SAPHIR.

*Line 133: do you mean uncertainty in a metrological sense? If not, you might want to change the word used. This is linked to the comment about lines 96-108.*

Yes indeed, we replaced "uncertainty" with "shape" in the sentence. It now reads:

**Lines 149 to 150:** "The averaged PDF encompasses all the available information of the reference RH such as the mean (first moment), spread, shape, and extremes of the distributions."

*Line 164: what is the uncertainty here? Source, magnitude? Or is it an error?*
This paragraph has been reworked and it now provides more details about the uncertainties in the reference dataset. We added a sentence explaining the simplification and our choice to keep a common uncertainty value representing all layers:

**Lines 179 to 181:** "The 15% RH uncertainty value is the smallest that allows to encompass the uncertainties of all pressure layers (see paragraph 2.1 or Brogniez et al, 2016 for a more complete analysis of uncertainty)."

*Line 192: I don't think you mentioned what you're a priori error assumption is before this point, what is it? Do you get an a-posteriori error? Do you calculate the error reduction?*
We do not use an a priori error nor an a posteriori error. The term "a priori" is used to state that in the deterministic comparison there is an assumption made on the uncertainty and based on the characteristics of the retrieval scheme. In other words, assuming "a priori" that the error is Gaussian yields to bias the analysis.

*Figure 5: Did 12:00 UTC look different? Is there any correlation to convection?*
Each case shows different patterns but yes, there seems to be a correlation between convection and the patterns in the model's biases.

**3. Technical Comments**

*Line 17: ".The study first ..." – change to ". This study first ..."*
Changed in the reviewed manuscript.

*Line 18: "It warrants the need ..." - this sounds like you are eluding to a future direction in a conclusion. Would something more like "We demonstrate the need ..."*
Changed in the reviewed manuscript.

*Line 33: change "relies" to "rely"*
Subject "The accuracy" is singular. "relies" stays as-is.

*Line 72: "such a probabilistic approach." – missing 'a'*
Changed in the reviewed manuscript.

*Figure 1b: X axis label missing, also cannot see bars for values > 10, log scale might help here*
The figures axis missing has been rectified and we used log scale in the updated figure. Figure 1b has been changed as follows:
**Figure 1b:**

[Figure]

*Line 137: "complementarities" – change to similarities*
Changed in the reviewed manuscript.

*Lines 232-236: need a space between %RH, i.e. % RH. There is no need for a space between the value and the percent, e.g. 12% RH.*
Thank you for the precision, this and all the following occurrences were modified.

*Line 265: need a space between %RH, i.e. % RH*

*Line 280: need a space between %RH, i.e. % RH*

*Line 295: need a space between %RH, i.e. % RH*

*Figure 6: need a space between %RH, i.e. % RH*

*Figure 7: need a space between %RH, i.e. % RH*

*Lines 343-359: need a space between %RH, i.e. % RH*

*Line 412: need a space between %RH, i.e. % RH*

*Line 427: need a space between %RH, i.e. % RH*

*Lines 439-440: need a space between %RH, i.e. % RH*

---

## Author Comment (AC2)

We would like to thank the anonymous reviewers for their very detailed comments on our work. The two reviewers have highlighted important imprecisions in the paper. We hope that our answers and the modifications added to the manuscript will help future readers better understand our work and avoid any misinterpretation.

This document presents our answers to RC1. For added clarity, we colour coded our answers as follows:

**Black: original comments from Anonymous Reviewer #1**
Blue: our direct answer
Green: corrections and/or additions as they appear in the updated manuscript. The new line number also appear.

**GENERAL COMMENTS**
=================
*The paper describes a new method to compare satellite measurements against model data leveraging inter quartile ranges derived from probability density functions. The method is well introduced and explained and employed to compare SAPHIR measurements against ARPEGE model data. It reveals significant and insightful differences and discusses these in detail. The topic fits the journal. The major short coming of this paper is the reference method chosen to be compared against the newly developed method. As demonstrated by the paper itself, it is not useful at all. The paper itself mentions the common use of second moments for such comparisons but uses a blanket, unmotivated 15% fixed error range itself. I recommend a major revision of the paper with a more commonly used reference method e.g., one employing second moments (standard deviations).*

**MAJOR COMMENTS**
==============
*This paper uses a very simple, so called "deterministic" method as reference. The method assumes a blanket +-15% error range as acceptable, independent of the actual level 2 data quality. This reference method is not properly motivated by the paper. Also, often dominant errors in radiative transfer inverse problems are of a multiplicative nature, which would affect high and low RH values very differently. This fact alone makes a constant error range an unrealistic assumption. The analysis of the paper itself suggests that a smaller assumed range might be more suitable. A very common method would be to use the standard deviation supplied by the data set (or at least compute it from the available distributions, if not given directly); such a method has obvious short comings, particularly for non-negative quantities, but is an "industry standard". The paper must use a more reasonable reference method to compare against or demonstrate that a blanket assumption of +-15\% fixed offset error is a widely used method.*

We understand the point raised by the reviewer. Below this choice is explained in depth in the answers to specific comments. The choice of methods is now better justified, and we believe that it will improve the manuscript.

**SPECIFIC COMMENTS**
=================
**line 43**
* * *
*While it is true that the forward model introduces uncertainty into the comparison in measurement space, almost all inversion schemes make use of a forward model (at least for training a statistical model with obvious implications). Due to the ill-posedness*

*of the inversion, this implies that the uncertainty in geophysical space is almost always larger than the uncertainty in measurement space, particularly as the representation in geophysical space might contain a large "null space" inaccessible to the inversion (e.g. high frequency vertical oscillations in temperature to nadir sounders). Thus, large discrepancies in geophysical space might be very small in measurement space. This is one of the reasons, why assimilation prefers assimilating radiances in contrast to geophysical quantities (which are much easier to assimilate). The current text reads as if comparing in measurement space would be disadvantageous, while a very strong case can be made for the opposite. A large disadvantage in comparing in measurement space is that it is much more difficult to identify the reason for a disagreement in geophysical space and thus the "faulty" model quantity.*

We agree with the reviewer's comment and think that the paragraph may indeed induce confusion. The purpose of the sentence in line 43 is to demonstrate that each and every comparison method, both in geophysical and observation spaces, is associated with uncertainties. We reworked the paragraph to avoid any misunderstanding. It now appears as:

**Lines 37 to 46:** "The comparison between remote sensing and numerical simulations may be performed either in the geophysical or observation space, each one being associated with their own uncertainties. In the geophysical space, the model geophysical variables are evaluated directly against remote sensing estimations based on a retrieval scheme. This retrieval scheme can be an inversion algorithm that is usually developed from a dataset that is supposed to fully encompass the atmospheric variability and the physics of the problem, or from an optimal estimation method built from global climatologies (see for instance Rosenkranz, 2001; Lerner et al, 2002; Karbou et al, 2005; Divakarla et al, 2014). In the observation space (e.g., radiance), a forward model is used to convert the simulated atmosphere into synthetic remote sensing measurements (Morcrette, 1991; Soden and Bretherton, 1994; Brogniez et al., 2005; Chepfer et al., 2008; Bodas-Salcedo et al., 2011; Jiang et al., 2012; Tian et al., 2013; Steiner et al., 2018). This model-to-satellite approach relies on the accuracy of the forward model to simulate remote sensing observations for a given atmospheric state (Weng, 2007), while strong uncertainties may remain (Geer and Baordo, 2014; Brogniez et al., 2016) and challenge the diagnosis of disagreement in the geophysical space."

**References:**
Rosenkranz P. 2001. Retrieval of temperature andmoisture profiles from AMSU-A and AMSU-B measurements. IEEE Trans. Geosci. Remote Sens. 39: 2429–2435.
Lerner, J. A., Weisz, E., and Kirchengast, G., Temperature and humidity retrieval from simulated Infrared Atmospheric Sounding Interferometer (IASI) measurements, *J. Geophys. Res.*, 107( D14), doi:10.1029/2001JD900254, 2002.
Karbou F, Aires F, Prigent C, Eymard L. 2005. Potential of Advanced Microwaves Sounding Unit-A (AMSU-A) and AMSU-B measurements for atmospheric temperature and humidity profiling over land. J. Geophys. Res. 110: D07109, DOI: 10.1029/2004JD005318.
Divakarla, M., et al. (2014), The CrIMSS EDR Algorithm: Characterization, Optimization, and Validation, *J. Geophys. Res. Atmos.*, 119, 4953– 4977, doi:10.1002/2013JD020438.

*line 103*
* * *
*Please provide an introduction to "beta probability density functions". The references in the vicinity do not explain the term. A mathematical beta distribution has two free parameters, which seems in principle feasible to derive for six layers from six BT measurements including error estimates. I do not believe that most readers are familiar with the term such that it deserves a better introduction, especially as it seems to lay the foundation for the latter IQR method. Also, one would derive by multivariate regression, under Gaussian assumptions, a maximum likelihood vector and a*

*covariance matrix detailing correlation in the data (optimal estimation). Typically, the weighting functions of the sounder are not sharp enough to neglect correlations...? Either way, please introduce the satellite level 2 product and its supplied diagnostics/error terms in sufficient detail.*

Thanks for the suggestion. We have modified the paragraph to better introduce the beta distribution and explain its usage, as follows:

**Lines 106 to 115:** "The retrieval of RH profiles is based on a multivariate regression scheme that provides the parameters (α, β) of a Beta probability density function of the estimated RH alongside the mean and standard deviation. This regression scheme is applied to every footprint and every pressure layer. The Beta distribution is chosen over a more classical gaussian model for its ability to better account for the spread and asymmetry around the mean that is more adapted to the study of atmospheric RH (see also the discussion in Stevens et al. 2017), while also representing the uncertainty of the retrieval scheme and the radiometric noise. The Beta model is used as follows:

$$PDF_{FS}(RH; \alpha, \beta) = \frac{RH^{\alpha-1}(1-RH)^{\beta-1}}{\int_0^1 u^{\alpha-1}(1-u)^{\beta-1}du} \ , (\alpha; \beta) > 0 \tag{1}$$

With $PDF_{FS}(RH)$ is the probability density function of $RH$ defined on the interval [0;1], (α, β) are the parameters of the distribution. The subscript "FS" stands for SAPHIR's "Footprint Scale".

*line 131*
*--------*
*Is the averaged PDF retained, which can be a rather arbitrary function (discretized in some fashion, I assume), or is effectively only mean and sigma or the IQR computed? The example PDF look very Gaussian-like in all cases and suggest such an interpretation. If the actual shapes are different, maybe some PDFs in the visualization should look more "wild".*

Since the SAPHIR retrievals at the footprint scale are not gaussian (see answer to comment on line 215 hereafter), only retaining the mean and sigma is not sufficient to represent the complexity of the estimations. The PDF averaged over each 0.25° x 0.25° gridbox of the ARPEGE model and for each atmospheric layer are discretized into 101 bins of 1%RH. We have replaced former Figure 1a with the figure below that showcases some wilder pdfs and an overall wider range of shapes to better illustrate the variability of cases found in the data set.
**New figure 1a:**

[Figure]

In addition, to avoid confusion and emphasise the change of scale, we refered to the probability density functions retrieved by SAPHIR at the footprint scale as "PDF$_{FS}$", with a subscript "FS"

standing for "Footprint Scale", and kept "PDF" without any subscript for the averaged PDF of the 0.25° x0.25° grid that were manipulated throughout our work. The manuscript has been updated to include this notation. We also explained further the change of scale by adding the following paragraph:

**Lines 144 to 149:** "For each model gridbox and each atmospheric layer, all the footprints' $PDF_{FS}$ are averaged together to compute an unconditional distribution of the RH averaged at the ARPEGE scale, as follows:

$$PDF(PDF_{FS}) = \sum_{i=1}^{N} PDF_{FS}(x)_i \times \frac{1}{N} \qquad (2)$$

With $PDF_{FS}(x)$ being the individual footprint scale distributions and $N$ the number of footprints collocated within the gridbox. The averaged PDF is discretized into 101 bins of 1%RH. It encompasses all the available information of the reference RH such as the mean (first moment), uncertainty, and extremes of the distribution."

*line 163*
*--------*
*The PDF suggests that a value of +-15% is too generous. Staying in a Gaussian framework this looks like a 2-sigma value, whereas the CDF based method with the CDF of 0.5 would correspond to being in an interval of even less than +-1 sigma (being within one sigma has a probability of 68\%). The proposed method is sound, but the chosen example seems very biased. Even without using arbitrary PDF functions, a Gaussian approximation and error analysis should be able to provide better results than shown. Only if the PDF/CDF are non-Gaussian, an improvement will be achieved. To that end, the authors should demonstrate the difference to the (too) common Gaussian distribution assumption is significant.*

As for the previous figure, the PDF shown on Figure 2 is from a gaussian-like case for the simple reason of showing an illustration that is not overloaded. This particular PDF-CDF and associated simulated value are from the comparison database. It is part of the 99.99% of the dataset that does not pass the normality test of Shapiro-Wilks. This point is developed in our answer to next comment (line 215).
While the +-15% RH may seem too generous in this case, in other cases it would seem too strict. While every case is different, this example clearly illustrates that sometimes the model forecast may be close to the mean estimated value but still fall into the extreme quartiles of the probability distribution. The value of 15% RH has been chosen following the work presented in Brogniez et al, 2015: this is the smallest value allowing to encompass all layers' uncertainties. Please refer to comment on line 219 for a more complete answer on that matter.

*line 215*
*--------*
*The interquartile range as central concept deserves at a one-sentence explanation in addition to the back-reference. Using an IQR instead of the PDF loses a lot of information as it boils the arbitrary shape down to two simple numbers, comparable to the Gaussian approach with mean/sigma. I do not expect large differences unless strange, e.g., bimodal distributions, distribution appear. What is here the experience of the authors?*

As rightly suggested, a sentence developing the concept of IQR has been added as follows:
**Line 237:** "The IQR represents the difference between the RH values corresponding to probabilities of 0.75 and 0.25".

We agree that the IQR only keeps part of the information contained in the distribution. The probability $P$ that indicates the position of $RH_{mod}$ within the distribution of $RH_{obs}$ complements

the analysis. It is also justified as a significant number of non-gaussian and even multi-modal distributions were observed. We conducted a Shapiro-Wilks test ($α_{SW}$ = 0.01) on all the SAPHIR dataset to test the PDFs for normality. This test renders a p-value for each PDF. If this p-value is found above $α_{SW}$, it would play in favour of the hypothesis of a normal distribution. If the p-value is lower than $α_{SW}$, the hypothesis is null and the sample is ruled out as non-gaussian. We found an overwhelming majority of non-normal cases (> 99%), and this regardless of the atmospheric layer and timestep. To reinforce the discussion in the paper, we have added a short description of the test and its results as follows:

**Lines 231 to 235:** "However, a Gaussian model would not have been adapted to the dataset. A Shapiro-Wilks test is run with each and every individual PDF of the data set (with $α_{SW}$ = 0.01). The Shapiro-Wilks test is a widely used test of normality in statistics (Shapiro and Wilks, 1965). Finding a p-value above $α_{SW}$ would mean that the null hypothesis ("the PDF fits a normal distribution") cannot be refuted. For each atmospheric pressure layer, more than 99.99% of the PDFs have p-values under $α_{SW}$ meaning that almost none of them can be qualified as gaussian."

We have added hereafter some striking examples picked within the dataset that show some wilder, multimodal PDFs (blue curves) along their corresponding normal distribution drawn using the same mean and standard deviation (red curves). Those are only presented for the sake of the discussion but are not included in the manuscript.

[Figure]

*line 219*
* * *
*Again the 15% uncertainty come up. The authors make a compelling argument against Gaussian models, but picking a fixed 15% uncertainty is much worse than a simple Gaussian model-based uncertainty estimation would be. With a deterministic uncertainty of, say, 30%, the proposed method would compare even more favourably. Please provide a reference to the chosen value of 15% being a reasonable error estimate for the level 2 product. Looking into some of the given references, I couldn't find it. Much better would be a comparison against a traditional Gaussian error analysis. The chosen confidence interval can be compared against being within some factor times sigma of the derived value.*

We fully understand the concern regarding the +-15% uncertainty. In Brogniez et al, 2016 (table 1), which evaluates the SAPHIR retrieval of RH using radiosoundings collected in the tropical regions, the uncertainty (standard deviation) of the retrieval is calculated for each atmospheric layer above ocean on the one hand and continent on the other hand. Overall, the values range from 3.6% to 15.8%. As high as this value may seem, 15% is the smallest value that allows to roughly apply the method regardless of the layer. As our work focuses on presenting the adaptability of the probabilistic approach, we think that having a more precise deterministic method does not add any weight to the analysis. We also demonstrated why the suggested traditional gaussian approach is not an appropriate methodology while answering to previous comments.

We corrected the uncertainty values and developed the paragraph related to the uncertainties of SAPHIR's retrieved profiles as follows:

**Lines 115 to 119:** "As detailed in Brogniez et al (2016, Table 1), the bulk standard errors of the dataset lie in the range [3.6-14.8] %RH, depending on the pressure range (3.6%RH for layer 250-350 hPa, 15.8%RH for layer 750-800 hPa). These have been estimated using oceanic and continental radiosoundings collocated with satellite overpasses. Stevens et al (2017) also highlighted the role of the vertical inhomogeneities in the discrepancies, strong gradients of moisture being the most difficult to capture by the passive sensors."

We added a sentence explaining the simplification and our choice to keep a common uncertainty value representing all layers:

**Lines 179 to 181:** "The 15% RH uncertainty value is the smallest that allows to encompass the uncertainties of all pressure layers (see paragraph 2.1 or Brogniez et al, 2016 for a more complete analysis of uncertainty)."

*Line 303*
*--------*
*Showing the individual distributions of RH_mod and RH_obs would be interesting as well. The Talagrand diagram suggests, as the authors note, that the model assume extreme values more often than the observation. The individual distributions should show the same in a maybe more accessible/familiar manner.*

[Figure]

Distributions of RH_mod vs RH_obs

As suggested, this figure shows the overlaying individual distributions of RH_mod and RH_obs. The histograms show the tendency of the model to predict more extreme values than

what is observed. However, we do not find that this figure adds anything in regards to figures 6a and 6b. While simpler to understand, this figure shows in our opinion a less striking difference between the model's simulations and the mean retrievals.

**Figure 8a**
*---------*
*The colour scale of this figure hides a lot of detail as can be seen by the fact that nearly everything is grey. Another indication that the +-15% assumption is not good. I bet a non-linear colour scale blowing up the currently grey part would reveal a lot of interesting details.*

Hereafter are three maps: (from top to bottom) the previous version, one showing the same results with a modified non-linear colour scale, and the probabilistic results (also as shown in the current paper) for comparison.

[Figure]

The aim of the comparison between the 1st and 3rd maps in the paper is to show the difference in spread of biased zones when using the probabilistic approach instead of the deterministic one. Of course, showing the biases lower than the 15%RH confidence interval adds details. Even when blowing up the areas with less-than-15%RH biases, the difference in contrast with the probabilistic results is striking. The probabilities $P_{3M}$ (bottom map) are most often in the extremes almost everywhere. Our method is a lot more precise than using a smaller deterministic confidence interval. We chose, and explained our motivations in previous answers, this fixed value of 15%RH and thus we have to be consistent within the results shown.

**line 426**
*--------*
*Almost all level 2 satellite products from nadir or limb sounders offer a second moment (standard deviation) as diagnostics, many go beyond that (covariance matrices, error*

*terms from different sources). The analysis suggests that a 15% error assumption is not reasonable for the current data set. Using a proper second moment instead would certainly deliver more useful results. The employed method uses the more useful IQR, which is likely superior to a more simple first/second moment consideration. This is, quite sadly, not demonstrated by the paper.*

Indeed, we are aware that a lot of satellite products offer a second moment. However, the propagation of uncertainties, as also put forth by Stengel et al, 2017 (ESSD), assume Gaussian distributions. As discussed above, our data do not follow the Gaussian assumption. We do believe, in this case, that using the IQR is much more appropriated than using second moment.

**Lines 448 to 450:** "Moreover, nowadays, a lot of satellite products offer second moment that enable inter comparison studies. However, the propagation of uncertainties assumes a Gaussian distribution which is not the case here. We developed a probabilistic approach for the retrieval of RH that releases such assumptions."

*MINOR REMARKS*

**═══════════**

Thank you for your vigilance! All undermentioned typos were corrected.

*line 36*
*-------*
*two way -> two ways*

*line 92*
*-------*
*km2 -> km²*

*line 211*
*--------*
*gaussian -> Gaussian*

*line 394*
*---------*
*P>0.25 -> P<0.25*

*Figure 9a*
*---------*
*The colour scale should reflect the three regimes that have so far been used, i.e. P< 0.25, 0.25<P<0.75 and 0.75 <P.*
We reworked the figure to limit the color palette to the three aforementioned intervals. It now appears:
**Figure 9a:**

[Figure]